# Analysing the Policy Delivery System and Effects on Territorial Disparities in Italy: The Mechanisms of Territorial Targeting in the EU Rural Development Programmes 2014–2020

**Francesco Mantino** [1,*], **Giovanna De Fano** [2] and **Gianluca Asaro** [2]

1   Council for Agricultural Research and Economics, Centre for Policy and Bioeconomy, 00187 Rome, Italy
2   National Research Council-IRET, 00010 Rome, Italy
*   Correspondence: francesco.mantino@crea.gov.it; Tel.: +39-06-47856430

**Abstract:** This study aims to answer the following research questions: (a) to what extent do EU rural development supports for investments address territorial differences of rural areas, especially concerning the differences between rich and intensive areas, on one hand, and marginal and peripheral rural areas on the other hand; (b) how does present governance and the delivery system of measures supporting rural investments contribute to the financial support of marginal and peripheral rural areas? To respond to these questions, the research examined 747 calls for tender in all Italian regions related to 16 types of investment measures and a global amount of EUR 67 billion Euros and 49,410 representative projects approved in 17 Italian regions during the period 2014–2020. Delivery mechanisms included the rules that have been set up to define recipient eligibility and selection criteria of the rural development programmes. The distributive effects of RDP investment support measures appear evidently uneven, especially in agricultural and agro-industrial competitiveness measures, which are mainly allocated in already dynamic and strongly competitive areas. Delivery mechanisms boost disparities when funds are allocated through the "open competition" approach. Instead, the modulation of territorial criteria in the implementation phase can provide effective results in terms of reducing disparities in fund allocation and outreaching the most lagging areas. There are two main novelties of this research: (a) the analysis of territorial criteria in the calls for tenders for investment support, and (b) the effects of these criteria on expenditure distribution at the municipal level (LAU2 in the EU nomenclature). This study has been carried out outside the formal methodological approaches promoted by the European Commission for RDP evaluation and might be considered a complementary approach to evaluation reporting activity. This study might provide two significant contributions to the debate on rural areas. First, a "combined" approach to the definition of rurality (mixing "structural" and "locational" approaches) might provide a better analytical framework in line with the evolution of the literature on rurality. Second, the delivery systems that put more emphasis on territorial targets, as they were presented in this study, might be an essential component of a place-based policy.

**Keywords:** rural development; rural policies; policy evaluation; EU common agricultural policy; rural areas; territorial impact assessment; policy delivery system

## 1. Introduction

The 2013 CAP reform (implemented and phased in during the 2014–2020 programming period) has had the primary aim of responding to the present challenges the EU was facing (both within agriculture and the wider context). These include economic challenges (food security, price stability, promoting productivity growth), environmental challenges (climate change, greenhouse gas emissions, habitat conservation, biodiversity, including climate change), and territorial challenges (vitality of rural areas, diversity in agriculture, rural resilience).

To address such diverse challenges, the 2014–2020 CAP was structured along three general objectives:

1.  Viable food production, with a focus on agricultural income, agricultural productivity, and price stability;
2.  Sustainable management of natural resources and climate action, with a focus on greenhouse gas emissions, biodiversity, soil, and water;
3.  Balanced territorial development (BTD), with a focus on rural employment, growth, and poverty in rural areas [1].

The three general objectives aligned, and fed into, the more general "Europe 2020" goals of smart, sustainable, and inclusive growth, which were translated into six priorities for the EU's rural development policy for 2014–2020, and eighteen corresponding focus areas [2]. The sixth priority is particularly significant for the subject of this study, as it deals with social inclusion and economic development.

BTD involves territorial cohesion while ensuring balanced and resilient growth across all EU regions. The EU rural development instruments particularly address depopulation/abandonment, remoteness and housing, access to research and innovation, social rights, and cultural heritage to name a few. More specifically, BTD refers to territorial cohesion, and convergence (a complementary policy objective), aiming to address the development gaps between economically flourishing regions/areas and those falling behind, through targeted policy interventions and investments. Further, it refers to "upward convergence", which is the policy aim to improve the working and living conditions and economic factors, of all Member States and regions. Upward convergence recognises the concept that closing the gap between regions/areas is not enough, and rather, all regions should experience an upward development trajectory.

In this context, this paper focuses the analysis of Italian Rural Development Programmes (RDPs) in the period 2014–2020 and aims to answer the following research questions:

(a)  To what extent does the EU rural development investment support address territorial differences of rural areas, especially those between rich and intensive areas, on the one hand, and marginal and peripheral rural areas, on the other?
(b)  How might the governance and the delivery system of the rural development measures contribute to the financial support of marginal and peripheral rural areas?

In this research, the focus is on policies supporting investments in rural areas. According to OECD New Rural Paradigm [1], the effectiveness of rural policies is strongly influenced by the financial share of investments in the total policy support, whereas income subsidies do not affect durable changes. For this reason, the analysis considers the main investment measures in RDPs.

Exploring these issues requires us to examine three strands of literature: (a) studies of the rural areas' diversity; (b) conceptual definitions of the delivery systems in policy analysis; and (c) methodological approaches framing the territorial impact assessment.

The definition of rural areas has changed over time, moving from purely locational or demographic to more complex approaches [2,3]. Several authors have identified a classification of the different approaches based on various indicators and the socio-economic relations between urban and rural areas. According to the results of the Horizon project called SHERPA [3], the approaches and methods to analyse and describe rural diversity can be grouped into six categories: (1) the administrative approach, based on legal-administrative data; (2) the morphological (or demographic) approach, which is one of the most used in the literature, based on the population characteristics (population size and density); (3) the location approach, based on the proximity of rural areas to urban centres; (4) the economic approach, based on variables such as agricultural GDP or the cost of services; (5) the landscape approach, based on land-cover and climatic conditions; and finally, (6) the combined approach, using a mix of the previous approaches.

Other authors comprise the different typological approaches to defining rural areas under three categories [4]: the "locational", "structural", and "combined" approaches.

The "locational" approach considers the proximity to a core urban area as a discriminant variable: territorial units are classified according to the time to commute to the central city. Similar territorial classifications have been adopted by the European Commission [5], the ESPON EDORA project [6] and other specific studies [4].

The "structural" approach considers a different set of criteria based on the percentage of inhabitants in small villages and the magnitude of the core urban centre in each region. These structural variables define three types of territorial units: predominantly urban, intermediate, and predominantly rural [7]. This urban–rural typology has been codified by EUROSTAT statistical reports at the level of NUTS3 regions [8]. A "structural" approach is also applied in other studies based on the economic activities composing the socio-economic structure of a given territory [4].

The "combined" approach usually mixes the criteria used in the "locational" and "structural" approaches [9,10]. This approach has been developed over time by OECD studies [11], which modified previous OECD typologies and brought about a new classification: intermediate areas close to a city, intermediate remote, predominantly rural areas close to a city, and predominantly rural remote. Nevertheless, this approach has been criticised since it does not allow us to understand the profound differences among rural areas [12]. Furthermore, a definition less dependent on the role of urban centres, more appropriate indicators and territorial scales are needed for policy design [13].

This evolution of the approaches to rural areas has not prevented the significant heterogeneity of rural classifications among countries due to the differences in geography, socio-economic conditions, and administrative traditions [14,15]. Furthermore, the literature on the classification of rurality, especially that using the OECD models, seems to neglect that rural areas have endogenous potentials and are not simply "the space between urban nodes" [16]. Moreover, the definition of "rural" does not include only the differences between rural and urban but also within rural areas [17]. Finally, the definition of rural areas proposed by supra-national organisations (OECD, European Commission, and EUROSTAT) inevitably overlaps with more policy-oriented definitions linked to the application of RDP measures (i.e., less developed areas, NATURA 2000 and high nature value areas, LEADER areas). This overlap contributed to weakening the potential effectiveness of the more general definitions.

The definition of rurality has practical implications for policy analysts since it allows an understanding of how the different EU policies and the CAP can generate diverse impacts on the typologies of rural areas. Policy impacts on rural areas can depend on the type of delivery model underpinning the distribution of policy support. In the highly diversified EU context, national or regional institutions and local agencies can implement the same policy instrument, which might encompass eligibility and selection rules quite different from region to region. Within the same country, we might observe significant differences in the delivery models, greatly depending on political choices and organisational systems [18]. The notion of a delivery system has been developed within the second strand of literature that is significant for this study.

This notion of a delivery system is relatively new and has been particularly explored in EU policies, given the complexity and the different levels of governance involved [18–20]. In the field of rural development policies, in particular, "*little practical research has compared rural development policy delivery mechanisms*" [19] (p. 157). Identifying the delivery system is a common objective of the current literature, but the approach can differ. Sandiford and Rossmiller [20] say that understanding the delivery system implies "*the system through which the policy is implemented to transfer the benefits from wheresoever they derive to the hands of the targeted recipients*" (p. 6), which, in turn, means exploring institutions with functional responsibilities, functions supposed to be performed, policy instruments and "*formal and informal rules, regulations and safeguards built into or assumed to be inherent in the system in the attempt to ensure that the benefits flow to the targeted recipients and to monitor the eligibility of the participants, and other compliance rules*" (p. 6). These authors include all these aspects in the structural analysis of the delivery system, which should be integrated by other two

components: conduct (how institutions operate to ensure that the system delivers the benefits to the recipients) and performance (how well the policy delivery system meets the original objectives of the policy and effectively reaches the targeted recipients). ENRD [19] and Mantino et al. [18] focus more closely on rural development delivery systems, through a series of European case studies, and analyse the system of institutions, rules and phases governing the implementation of the most important rural development measures. In these studies, the role of territorial targeting is marginal. ENRD [19] reports that "*in the programming phase territorial targeting (including eligibility conditions and selection criteria) has usually been disconnected from the definition of rural areas—even though rural areas have been defined at the strategy level*" (p. 159). The weakness of territorial targeting was due to the vagueness of targeting in the programme design. However, ENRD analysis was limited to the programmes and did not consider the call for tenders which contain more detailed and specific delivery rules. These calls for tenders are usually the official documents where eligibility conditions and selection criteria are concretely defined.

Understanding how policy delivery works and how rural actors respond to policy incentives remains crucial in analyses of territorial impact, which is the third strand of literature (point c above) particularly linked to the objective of this work. This issue is relevant in general policy assessment, particularly in the family of territorial impact assessment (TIA) and rural proofing studies, which have been increasingly developed in the last decade [21–24]. More specifically, impact assessment procedures envisage a series of steps (screening, scoping, assessing, evaluating) requiring the selection of appropriate territorial units, the quantification of potential impacts and the territorial distinction of impacts. During the assessment step, the analysis is "*based on the assumption that the territorial impact of a policy will largely be conditioned by the intrinsic territorial characteristics (physical, economic social and administrative) of different regions and localities* (Zonneveld & Waterhout 2009). *These tools require suitable (sub-national) territorial units for identifying impacts*" [23] (p. 45). The definition of the most appropriate territorial units and the knowledge of which policies are addressed to these units are key aspects of any TIA approach, regardless of the methodology applied. Addressing unbalancing effects or enhancing effects in favour of territorial cohesion might support the policy-making process and be important for many types of policy analysis, for instance, evaluations made by external experts. The territorial analysis is strongly based on the mixing of quantitative and qualitative information, including the experts' knowledge and their experience of previous policy tools and their implementation in a highly differentiated context. Thus, the analysis of the territorial distribution of existing policies and capabilities of the different rural areas to access/exploit the policy tools can strongly inform better policy decisions.

## 2. Materials and Methods

The methodology adopted in this study comprises four main steps:

(a) Analysis of the definition of rural areas used by the Italian RDPs in the period 2014–2020 and collection of data related to the main characteristics of rural areas.

(b) Definition of the logical framework within which territorial criteria are used by RDPs and quantitative analysis of territorial eligibility/selection criteria and their coherence with the importance of different rural areas.

(c) Gathering data concerning the calls for tenders and the projects funded after tender's completion. The data gathering allows the creation of two different databases: the first related to eligibility and selection criteria set in each tender call, and the second database including information related to the projects funded by each tender.

(d) Analysis of the main factors explaining the allocation of investment support in the most marginal and peripheral areas.

The definition of rural areas was based on two different classifications, which might fall under two of the main typologies already described in the analysis of literature. The first one is used by the Italian RDPs 2014–2020 into four different types: (a) urban and peri-urban areas; (b) intensive agriculture areas; (c) intermediate rural areas; (d) lagging rural areas.

This classification can be categorised under the above-mentioned "structural" approach. This classification has been followed by 21 regional RDPs, with some adjustments due to the need for a more detailed and discriminating definition of the "intermediate" typology. This approach provided all Italian regions with a common definition of rural areas and substantially reduced the high variability of rural definitions across Italy [25]. In the Italian RDPs, the most contrasting differences occur between urban, peri-urban and intensive agricultural areas, on one hand, and lagging rural areas, on the other hand. Figure 1 illustrates on the right-hand side the RDPs map of rural areas and in particular lagging rural areas (dark green in the figure) mostly occupying mountain and remote areas and a great part of southern regions.

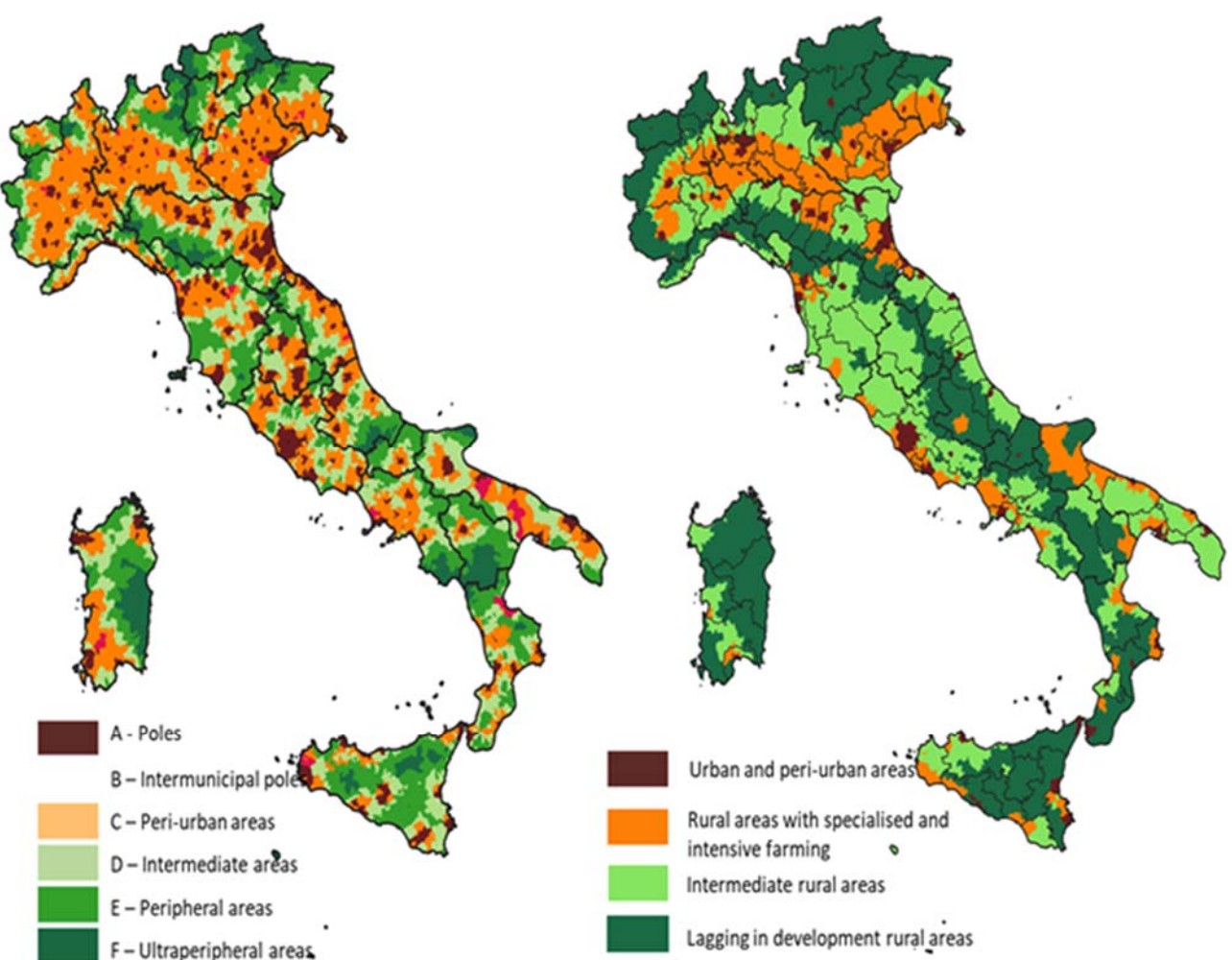

**Figure 1.** Official definition of rural areas used in Italian programmes of public intervention.

The same figure shows, on the left-hand side, the map of rural areas according to the definition of the national Inner Areas Strategy (IAS), which classifies rurality based on the "distance" from essential services (education, health and railways infrastructures). This second classification of the Italian rural areas can be comprised under the "locational" approach defined in the introduction. The IAS is a multi-fund national policy, which has been designed separately from the RDPs but under the same European framework (the, National, Partnership Agreement of European Structural and Investment Funds 2014–2020). The IAS definition of rurality has been used in Italy to identify peripheral and ultra-peripheral areas, and to focus on specific interventions to promote local development and provision of essential services in these areas [26]. According to this definition the most contrasting

disparities occur between poles and peri-urban areas on one hand, and peripheral and ultra-peripheral areas on the other hand.

The regional RDPs introduced further territorial definitions in the process of funds allocation, such as:

Inner areas (from the National Strategy for Inner Areas)

1.  Disadvantaged mountain areas;
2.  Areas with natural and specific constraints;
3.  Protected areas and NATURE 2000 zones;
4.  Rural areas included in the earthquake zone (in particular, in the Lazio region);
5.  Other criteria (i.e., based on the population size of municipalities).

Other minor definitions are drawn from the RDP legislation at the EU level. Disadvantaged mountain areas and areas with natural and specific constraints are definitions currently used in RDPs for the purposes of supporting areas characterised by natural and geographical handicaps, through public aid to low-income farmers (art. 31 and 32 of the EU Reg. 1305/2013). Areas falling under these definitions are used by managing authorities beyond their specific scope, as a possible territorial target for other regional RDP interventions. Protected areas and NATURE 2000 zones are other territorial definitions currently used in EU policies, including environmental and rural development policy measures, to address areas that are particularly vulnerable but rich in natural resources to be preserved. This analysis will use the Italian RDPs' definition of rural areas and IAS definition to compare the allocation of investment support at the territorial level, whereas all other mentioned definitions will be considered as not relevant for this purpose.

The RDP's delivery framework in Italy involves two main levels: programming and implementation. At the programming level (Figure 2), RDPs establish the first set of territorial priorities through the classification of rural areas and design of a strategy that set territorial targets. The rural area definition identifies the main differences between urban and rural, and diversities within the broad concept of rurality. The intervention strategy can set territorial targeting in different ways: either by allocating financial resources to some areas, or more often by choosing admissibility and selection criteria which give some priority to specific areas, based on socio-economic disadvantages, altitude (mountain areas), remoteness, etc.

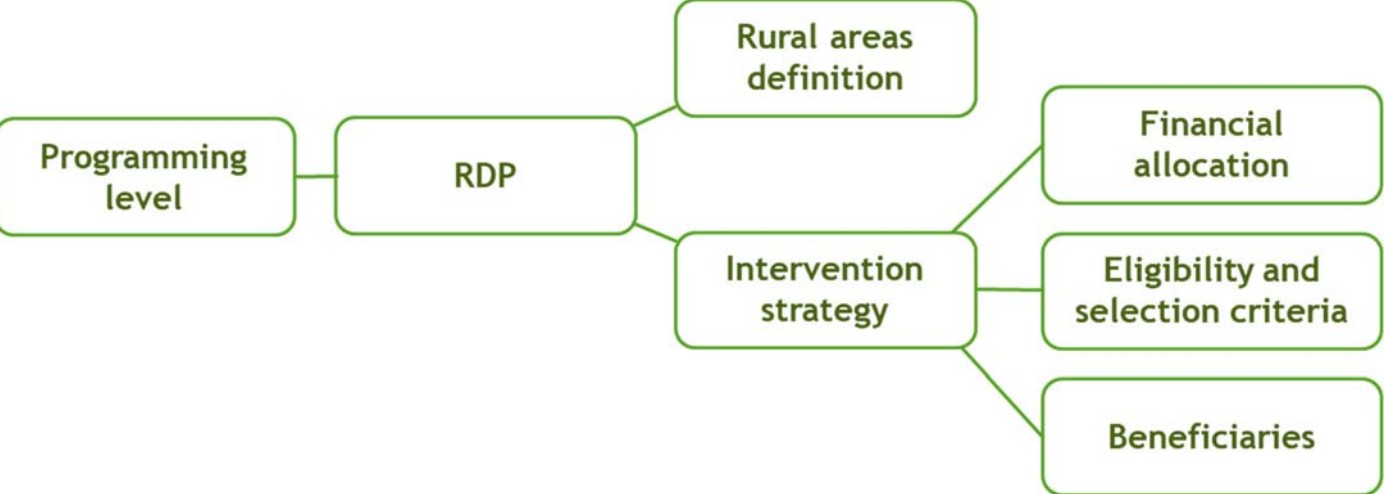

**Figure 2.** RDP programming level: early general definition of territorial priorities.

Regarding the implementation phase, RDPs influence territorial preferences by defining financial allocations, admissibility/selection criteria and types of beneficiaries (Figure 3).

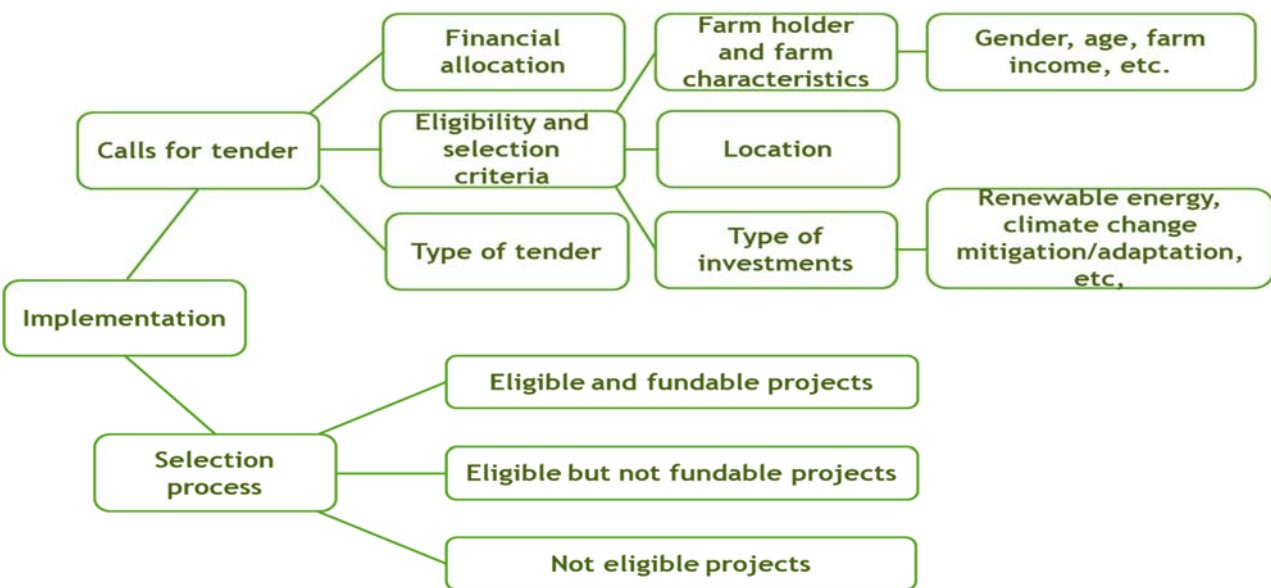

**Figure 3.** RDP implementation level (call for tenders and selection process): more specific definition of territorial priorities.

There are two subsequent phases. In the first one, the regional programme's managing authority (MA) defines more detailed implementing rules for each intervention measure in the tender procedure to prioritise the beneficiaries of policies. Regional MAs further tune financial allocation within the resource's availabilities set in the programme, and criteria to select beneficiaries. Furthermore, tenders are not all the same: some target specific areas or beneficiaries, and others are open to everyone regardless of their location and characteristics.

Eligibility and selection criteria are the most powerful and widespread means to target specific stakeholders. They can be discriminatory depending on the policymakers' preferences. For example, they can be addressed particularly to young entrepreneurs or farms located in mountain areas if analysis of territorial needs and the process of policy advocacy by rural stakeholders have exerted relevant pressures in this direction. This usually happens during the preparation of the tender call and even before that, in the process of the programme's design.

The second phase at the implementation level is the issue of the tender call and, once the tender deadline is closed, the selection process. MA selection commissions and local offices (usually sub-regional ones) assess and select the projects according to the scores set in the tender call. Each beneficiary project is assessed and scored following, among the others, territorial criteria. Based on these criteria, MA commissions approve a list of beneficiaries, which can be eligible and fundable, eligible but not fundable (usually because the beneficiary' project has got a lower score and projects with higher score exhausted the available funds), or not eligible (because the project has not reached a minimum score to be considered viable).

Table 1 presents the list of 16 investment support sub-measures we have considered in this study and the number of calls for tenders examined in all Italian regions. These sub-measures belong to four support measures:

1. Measure 4—Investment in physical assets;
2. Measure 6—Farm and business development;
3. Measure 7—Basic services and village renewal in rural areas;
4. Measure 8—Investments in forest area development and improvement of the viability of forests (only Sub-Measure 8.1—Afforestation/creation of woodland and Sub-Measure 8.6—Investments in forestry technologies and processing, mobilizing, and marketing of forest products).

**Table 1. The** RDP investment sub-measures, planned expenditures in 2014–2020 period and no. of calls for tenders examined.

| Sub-Measure | Measures Definition | Planned Expenditures (Million Euro) | Calls for Tender Examined (No.) |
|---|---|---|---|
| 4.1 | Investments in agricultural holdings | 27,005 | 164 |
| 4.2 | Investments in processing/marketing and/or development of agricultural products | 10,721 | 68 |
| 4.3 | Investments in infrastructure related to development, modernisation or adaptation of agriculture and forestry | 4762 | 66 |
| 4.4 | Non-productive investments linked to the achievement of agri-environment-climate objectives | 2686 | 66 |
| 6.1 | Business start-up aid for young farmers | 7889 | 74 |
| 6.2 | Business start-up aid for non-agricultural activities in rural areas | 625 | 12 |
| 6.4 | Investments in creation and development of non-agricultural activities | 4495 | 87 |
| 7.1 | Drawing up and updating of plans for the development of municipalities and villages in rural areas and their basic services and of protection and management plans relating to Natura 2000 sites and other areas of high nature value | 201 | 16 |
| 7.2 | Investments in the creation, improvement or expansion of all types of small-scale infrastructure, including investments in renewable energy and energy saving | 856 | 22 |
| 7.4 | Investments in the setting up, improvement or expansion of local basic services for the rural population including leisure and culture, and the related infrastructure | 974 | 22 |
| 7.5 | Investments for public use in recreational infrastructure, tourist information and small-scale tourism infrastructure | 596 | 20 |
| 7.6 | Studies/investments associated with the maintenance, restoration and upgrading of the cultural and natural heritage of villages, rural landscapes and high nature value sites | 885 | 33 |
| 7.7 | Investments targeting the relocation of activities and conversion of buildings or other facilities located inside or close to rural settlements, with a view to improving the quality of life or increasing the environmental performance of the settlement | 14 | 1 |
| 8.1 | Afforestation/creation of woodland | 1705 | 36 |
| 8.6 | Investments in forestry technologies and in processing, mobilising and marketing of forest products | 1633 | 51 |
| | Multi-measures calls | 1261 | 4 |
| | Total | 67,240 | 747 |

Source: authors' data base of calls for tender.

The study did not include the LEADER measure, for several reasons: in Italian RDPs, this measure is applied exclusively in lagging and intermediate areas and consequently is a territorial measure by definition; second, the financial share in regional RDPs is negligible compared to the total budget; third, LEADER would require, due to his peculiarity, a specific analysis. Due to these peculiarities and the presence of 194 Local Action Groups (LAGs) in Italy, LEADER needs to be object of a specific study with a similar methodology, to compare the differences between investment support managed at regional and local level (by LAGs).

Table 1 shows that we have examined 747 calls for tenders in Italian regions and a global budget of EUR 6,7 billion, representing more than one-third of the total planned expenditures for rural development support in Italy in the 2014–2020 period.

The rural development programming system in Italy comprises 21 RDPs, each with specific characteristics and definitions that complicate comparison. Indeed, the delivery of investment measures and the selection of investment projects at the regional level have been conditioned by the presence of some or more than one territorial definition. To analyse how territorial criteria have influenced the selection process, we collected quantitative

information on criteria and related weights used to rate projects applications from the Italian Rural Network database of calls for applications (see the Rete Rurale Nazionale website: https://polaris.crea.gov.it/psr_2014_2020/bandiPSR.htm (accessed on 31 August 2022)).

This study set up two different databases. The first database collected and systematised information regarding the calls for tender issued by the regional authorities in the application of RDPs. It was crucial to standardize data and try to thin out the differences among the calls for tender. The work was time-consuming since the required information had to be extracted from 747 calls for tender.

For all investment sub-measures and types of operations, the database was set up with codified information including the date of issue and project selection, financial allocation, eligibility criteria, and territorial selection criteria.

Data gathering included a comprehensive examination of all the criteria used by the RDPs with the ultimate objective of identifying the main tender typologies. Figure 4 shows that there are three different types of tender calls:

1.  Calls targeting specific areas (tender calls limited to a subset of rural areas of type A, B, C, and D or some other typology defined above).
2.  Calls accessible to all areas, but with some degree of positive discrimination for some specific areas (by using selection criteria). Figure 4 shows that these calls for tender represent the majority of total calls. In these calls for tenders, the highest scores are usually given to projects proposed by beneficiaries in lagging areas.
3.  Calls having no territorial discrimination and fully open to competition among rural areas.

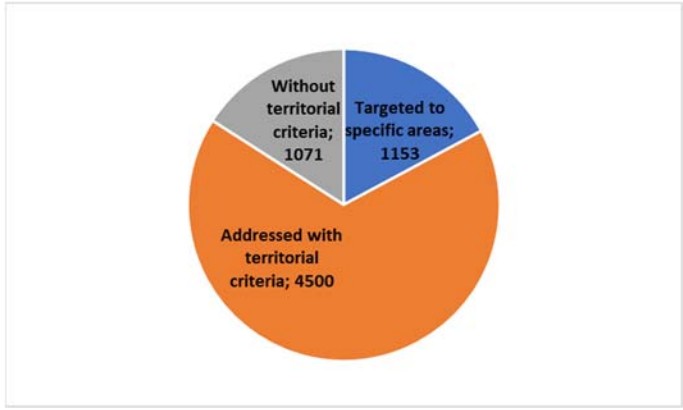

**Figure 4.** Calls for tenders according to the territorial targeting (expenditures in million EUR). Source: authors' data base of calls for tender.

Territorial selection criteria were weighted according to the score attributed by regional calls in the total score potentially given to the investment applications. This weight reflects the relative importance attributed by regional administrations to the different rural areas. In general, investments in lagging or mountain areas have the highest scores in absolute terms, but the rating in the total scores is highly variable from region to region. Figure 5 shows the average score of each territorial criterion, as calculated from the calls for tender in Italian RDPs for investment support. The highest average scores are attributed to investments in lagging in development rural areas (type D in the national nomenclature) and areas in the earthquake zone (in the Lazio region). The investment criteria were modulated according to the national definition of rural areas, by increasing the weighted score from type B (intensive agriculture) to type D. The priority given to Inner Areas is rather marginal.

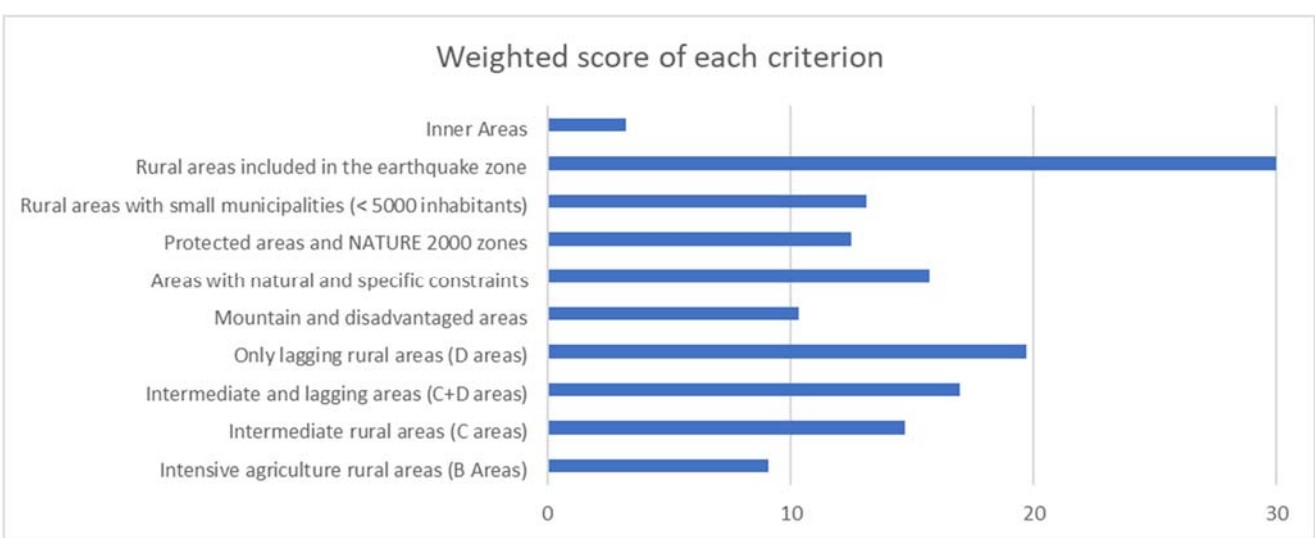

**Figure 5.** Weighted score of investment criteria by type of criterion. Source: authors' elaborations from the data base of calls for tender.

The creation of the call for tender's database was instrumental in developing a second database to test the effectiveness of territorial criteria set in the tender calls. This second database contains information on investment projects approved by the MA commissions and regional offices, partly derived from the first database regarding calls for tender characteristics, and partly gathered from the ranked lists of projects. By integrating the two sources of information (calls for tender and ranked projects), the second database includes:

1. Date of call for tender's issue and closure (drawn from the first database);
2. Location of investment (municipality-LAU2 level in the EU territorial nomenclature, which corresponds to the municipality-the lower level of territorial representation);
3. Eligibility and selection criteria (drawn from the first database);
4. Date of application approval;
5. Total investment (public + private quotas);
6. Total public (European + national) committed expenditures.

This database was set up with information drawn from the ranking lists published by regional administrations. Where ranking lists do not include the necessary information, missing data were integrated by the official database provided by the Ministry of Economy and Finance. Unfortunately, even this database is incomplete due to the lack of some regional expenditures. Despite these difficulties, the project sample was sufficiently representative of beneficiaries of RDP investment support measures: it consists of 49,410 projects approved in 17 Italian regions and including all 16 types of operations. Figure 6 compares in each investment measure the committed expenditures of the sample of projects approved with the planned expenditures in Italian RDPs. This comparison shows that the sample of projects approved is highly representative of measures considered, except for forest investments, due to the low rate of advancement across regions for this category of investment.

The project's sample is also representative of the main geographical regions of the country (North, Centre, South Italy). Regarding the types of tenders, most projects have been selected through ordinary call for tender, which implies calls for individual applications related to single measures (mono-measure applications). Other types of calls include multi-measure calls for tender (i.e., packages of measures such as agri-environmental agreements, young farmers' packages, sectoral and territorial integrated packages, etc.), all involving groups/partnerships of beneficiaries presenting collective applications. These multi-measure tenders represent a peculiarity of the Italian delivery model in the European panorama, widespread across regions but financially limited in the total applications.

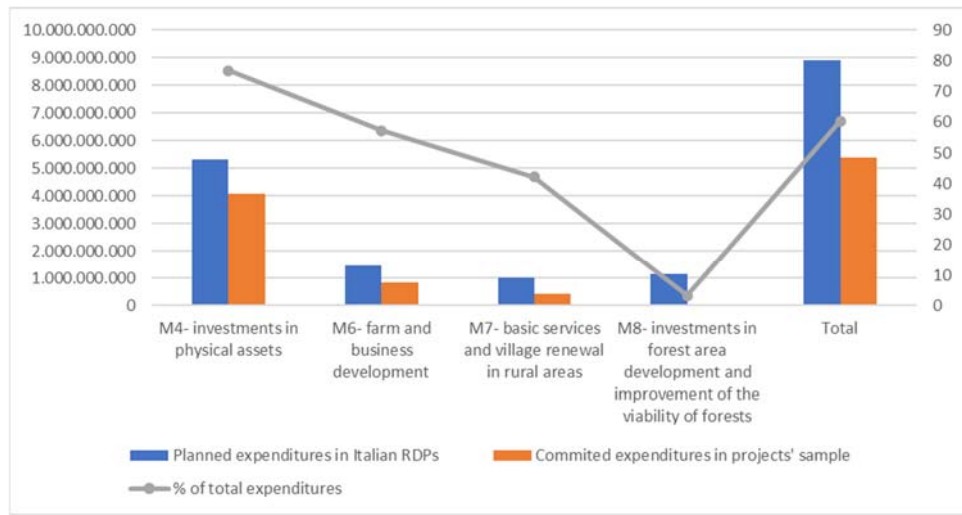

**Figure 6.** Investment committed expenditures (EUR), compared to RDPs' planned expenditures. Source: authors' elaborations from the data base of funded investment projects.

## 3. Results

This section is divided into two subheadings. The first describes the general distribution of investment support between peripheral/lagging areas and the rich and intensive areas. The second one aims to explore what is the role of the delivery system in generating this uneven funds' distribution.

### 3.1. Distribution of Investment Support between Peripheral and Non-Peripheral Rural Areas

To explore the investment allocation by type of rural area, the analysis makes a comparison of the two institutional definitions of rurality in Italy, according to the RDP and the Inner Areas' Strategy. Public investments supporting the beneficiaries' applications are the total committed expenditures 2014–2020, as surveyed until 31 May 2022.

Figure 7 illustrates the distribution of the financial expenditures allocated to projects in the two different types of areas, by region. According to both institutional definitions, the northern regions support investments in poles, peri-urban and intensive agricultural areas, i.e., the most dynamic areas. In central regions, according to the RDP typology, there are no significant differences, whereas according to the IAS, the pole and peri-urban areas receive the major share of funds. In the southern regions, results are more controversial since the two institutional definitions provide contrasting distributions: in the RDP definition, lagging areas receive more funds, whereas for the IAS, it is the other way round.

These different fund allocations may be influenced by the size of geographical areas and the share of different rural areas in each geographical context. To eliminate the influence of this factor, the funding allocation was "corrected" by the total agricultural area (TAA) in each region, i.e., the % of fund allocation was divided by the share of TAA, to obtain a ratio that can be less than 1 (the area gets fewer funds than its agricultural importance), or more than 1 (the area gets more funds than its agricultural importance).

Figure 8 shows the financial allocation of investment support when funds are "corrected" by TAA: when the size of the region is considered, regardless of the institutional definition, the richest areas benefit from a fund allocation, which is disproportionate compared with their agricultural weight. This disparity is clear in northern regions but is particularly evident in southern regions where structural and social weakness are more relevant than elsewhere. Lagging or peripheral areas never reach a fund allocation adequate to their agricultural role in the different territories of the country, except for in Central Italy, according to IAS definition.

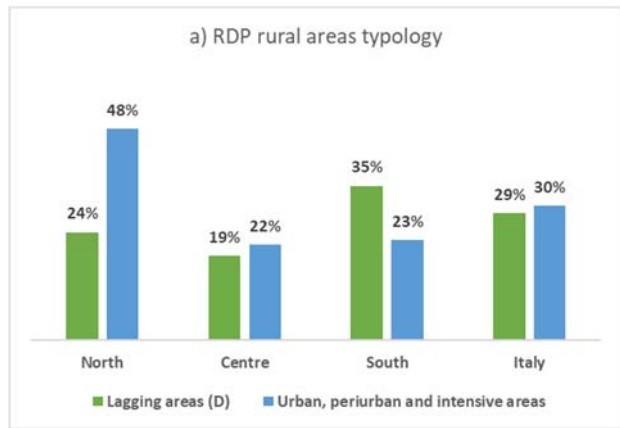
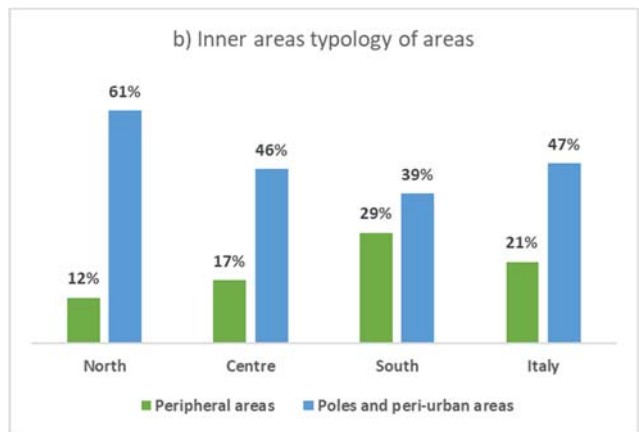

**Figure 7.** Percentage of total committed expenditures in Italy allocated by types of areas according to the two institutional typologies. Definition (**a**): RDP rural areas; definition (**b**): IAS rural areas. Source: authors' elaborations from the data base of funded investment projects.

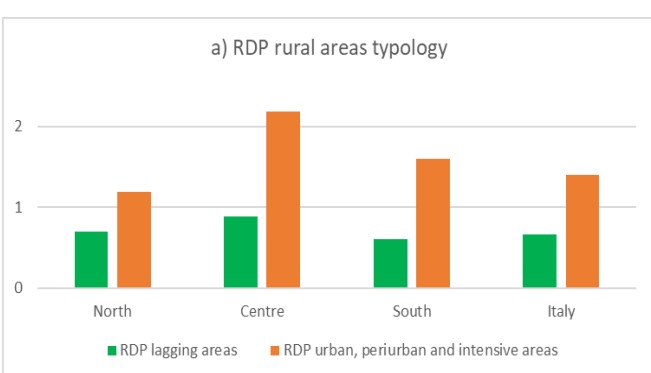
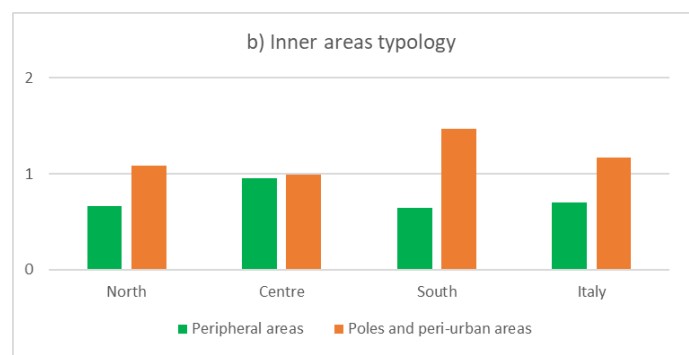

**Figure 8.** "Corrected" committed expenditures allocated by types of areas according to the two institutional typologies. Definition (**a**): RDP rural areas; definition (**b**): IAS rural areas. Source: authors' elaborations from the data base of funded investment projects.

The analysis of investment categories makes it evident that there are similar allocation patterns in different geographical contexts. In both territorial typologies, projects aiming to promote sectoral competitiveness (farm and processing/marketing investments, M4.1 and M4.2) are generously funded in more intensive and specialized areas (Figure 9). Other types of investments, such as infrastructures, non-productive investments (especially in the environmental field), small-scale rural services and, to some extent, non-agricultural investments (in tourism, leisure activities, etc.), are more heavily supported in peripheral or lagging areas. Investments by young farmers seem more balanced among the different types of areas and a major share of project funds is also addressed to intermediate areas. Contrary to expectations, most forestry projects are supported in the most intensive or intermediate rural areas.

This analysis may allow us to deduct that the investment support from the Italian rural development policies, as implemented at the regional level, responds to different demands from diverse rural areas. Rural areas express demands for investments which derives partly from their needs and partly from their capability to compete in funds' allocation. However, this allocation is strongly influenced by the type of delivery system that is set up at the regional level. For this reason, the analysis requires digging more deeply into the nature of the delivery systems that regions have set up to implement their policy instruments.

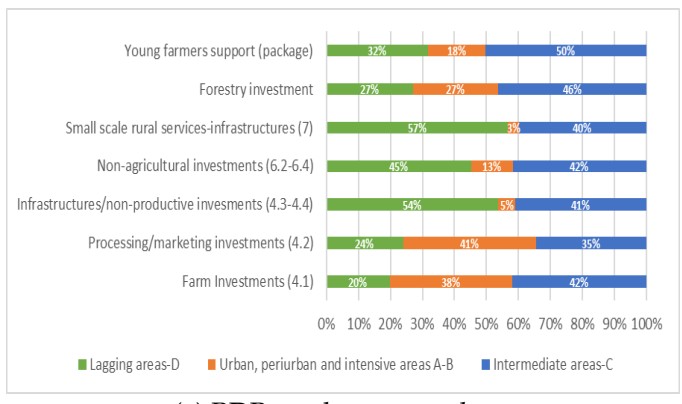

(a) RDP rural areas typology

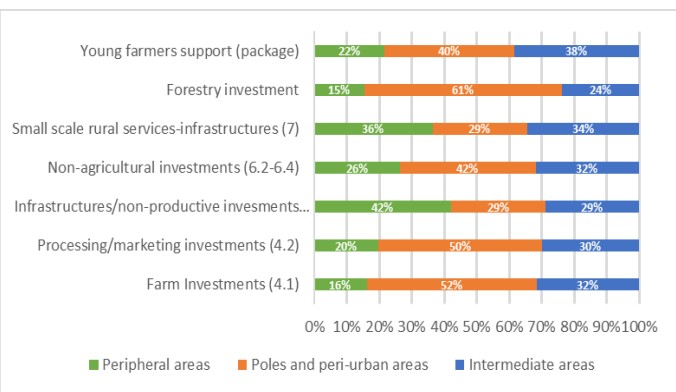

(b) Inner areas typology

**Figure 9.** Percentage of total committed expenditures by RDP rural areas and type of investment (RDP and IAS). Source: authors' elaborations from the data base of funded investment projects.

*3.2. How the Policy Delivery Mechanisms Can Increase the Territorial Disparities*

The criteria for assessing projects' eligibility and compliance with territorial priorities are decisive in determining the access to funds. They frame the concrete opportunities for beneficiaries in the different rural contexts. Table 2 shows the range of opportunities open to the different rural areas by considering combinations of eligibility conditions and territorial criteria in RDP selection procedures.

**Table 2.** Modes of project selection, based on combination of territorial eligibility and selection criteria.

| Presence/Absence of Territorial Criteria | | Territorial Eligibility. RDP Tenders Reserve Funds Only to: | | | | |
|---|---|---|---|---|---|---|
| | | All Areas | Intermediate and Lagging Areas (C + D) | Only Lagging Areas (D) and Inner Areas | Only Plains and Irrigated Areas (Type B) | Protected—NATURA 2000-HNV |
| Selection criteria for scoring projects | no territorial criteria | Competition among areas | Limited competition | Targeting lagging areas | Targeting non lagging areas | Targeting natural areas |
| | at least one territorial criterion | Mitigated competition | | | | |

Source: authors' classification.

Starting from the left-hand side, a situation of "competition among areas" is created when all areas are eligible for funds and no territorial criterion is set for potential applicants. When some territorial criteria are introduced and competition among potential beneficiaries is somewhat counterbalanced by criteria prioritising the most disadvantaged areas, there is "mitigated competition". Moving from left to right, increasing modalities of targeting specific rural areas are observed. For example, the combination of eligibility limited to some areas and territorial criteria identifies forms of "limited competition". Eligibility to funds, i.e., under the form of a ring-fencing mechanism, can be further limited to lagging, non-lagging or natural areas [3]. In some of these cases, fund reserve can be joined to territorial selection criteria. Territorial criteria can be different, as mentioned before: for the sake of simplicity, all criteria related to inner areas, mountains and disadvantaged areas have been grouped into one category of "lagging areas".

Among the modes of selection, "open competition" represents a significant share (Figure 10), but the most frequent mode is "mitigated competition", since regions generally include some territorial criteria in the selection process. Territorial targeting, whatever the rural area, allocates less than one-fourth of total expenditures. Tenders exclusively targeting lagging areas attract a negligible share of total resources and, paradoxically, receive the same amount of funds as non-lagging areas. This focus on the modes of selection makes clear how unfair the competition between areas in different structural and socio-economic

conditions is. However, to understand the implications of the different selection modes, it needs to be explored what results are achieved.

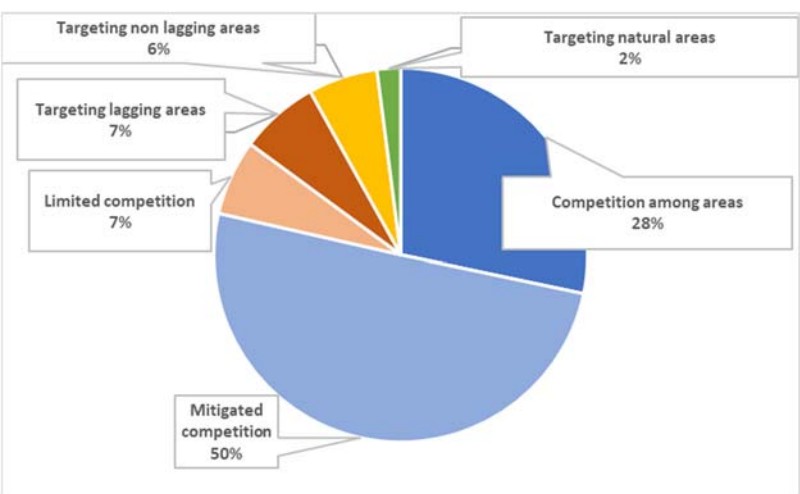

**Figure 10.** Distribution of committed expenditures by different modes of project selection (million EUR and %). Source: authors' elaborations from the date base of funded investment projects.

The mechanisms of access to funds change according to the policy instrument. Farm and agri-food competitiveness support mainly relies on "open competition" and "mitigated competition": these mechanisms are mainly applied in call for tender related to farm and processing/marketing investments (Figure 11). Young farmers' support is almost entirely addressed by "mitigated competition" calls, to allow access to funds for young farmers in the most disadvantaged areas. On the other hand, tenders with "limited competition" or targeting specific areas, in particular lagging ones, represent the prevailing mechanism for accessing non-agricultural investments, small-scale rural services, social infrastructures, and non-productive investments. These mechanisms provide a sort of fund reserve for lagging areas and explain why these kinds of investment support prevail in these areas.

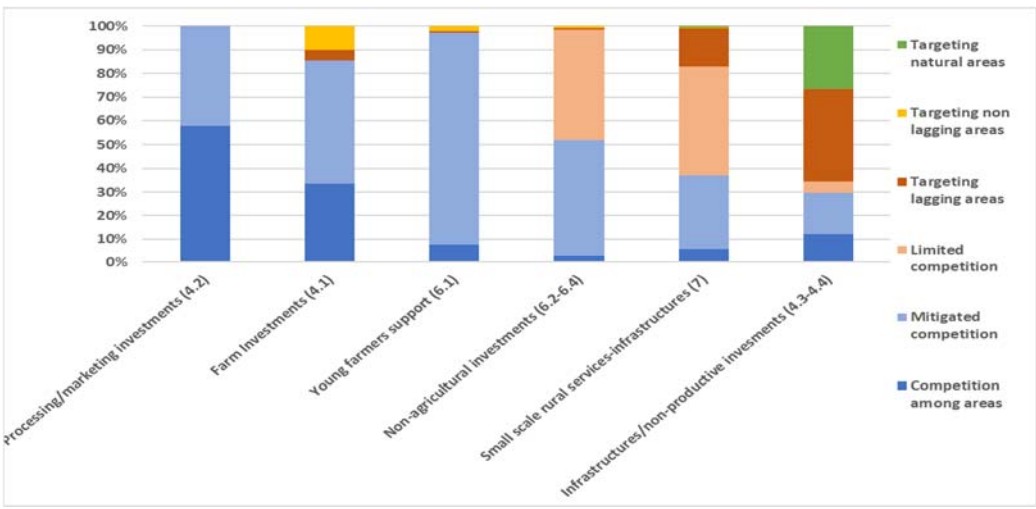

**Figure 11.** Distribution of committed expenditures by different modes of project selection in each type of investment support. Source: authors' elaborations from the data base of funded investment projects.

"Mitigated competition" deserves a particular focus, due to the widespread use of this mechanism in the Italian regions. In this call for tender mode, the scoring system (the intensity of score attributed to lagging areas) seems crucial in determining the scale

of territorial prioritisation. Scores are very variable across regions and even within each region across measures; they can vary from zero to thirty points and beyond. The higher the territorial scores in each tender, the higher the probability for a beneficiary in a lagging area to pass the selection process.

Figure 12 shows how access to funds in different rural areas changes as regions give increasing scores (out of total scores attributed to all criteria) to lagging areas in RDP call for tender. Moving from zero to more than 30 scores facilitates access for an increasing share of beneficiaries in lagging areas. This trend is evident until a score of thirty is reached, but the lagging areas' share remains however relatively high. Within the 15–30 score range, the share of committed expenditures in lagging areas reaches 40% of total commitments, which is twice as much as these areas would receive if the score was zero. It is worth noticing that a similar trend can be observed in expenditures for inner areas. This seems to confirm that an increasing prioritisation of lagging areas in rural development policy implies benefits even for the national Strategy for Inner Areas, funded by other European Funds. On the other hands, the share of expenditures in the most dynamic areas decreases from 50% (the situation with no territorial criteria) to 13% (the situation with a 15–30 score range).

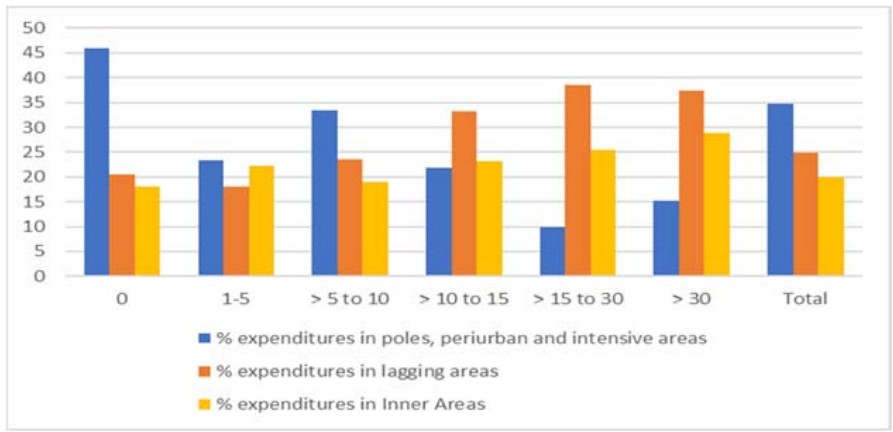

**Figure 12.** "Mitigated competition tenders": committed expenditures among different areas by range of scores to lagging areas. Source: authors' elaborations from the data base of funded investment projects.

## 4. Discussion

The analysis of delivery mechanisms allocating European Union and national funds has proved to be a crucial step in understanding the territorial impacts of rural development policies. Rural areas are highly diverse, in the European and national contexts since deep disparities occur between regions and within the same region. Policies that are territorially blind, implemented by assuming that all areas can fairly compete with each other and have the same opportunities of accessing funds, inevitably exacerbate these disparities over time [27].

Delivery mechanisms include, among other things, the rules which have been set up to define eligibility and selection criteria for the recipient of Italian rural development policy in the 2014–2020 period. These rules have been formally designed, although in a very general way, in RDPs, but their complete and operational definition appears in calls for tenders. Tender calls contain more detailed delivery rules than in RDPs and define the concrete opportunities for accessing funds by the potential rural beneficiaries. The first novelty of this research is the analysis of territorial criteria in the calls for tenders for investment support. Territorial criteria may represent a powerful tool to implement policies more targeted to specific areas: previous studies [27,28] have focused their analysis on targeting in RDP strategies, whereas the process of targeting is designed in regional calls for tenders.

The second novelty deals with exploring the effects of territorial criteria on the expenditure's distribution at the appropriate scale. Regarding this point, previous studies have analysed territorial policy effects at the NUTS3 level [28–32], which in the Italian case has to be considered too wide to represent territorial differences. As some authors pointed out [30] (pp. 1–2), "*a too aggregated scale of analysis not only implies a loss of information but may also incur misleading conclusions whenever relevant heterogeneity occurs at the smaller (i.e., more "local") scale*". This heterogeneity occurs at the NUTS3 level as this scale encompasses diverse typologies of rural areas. For this reason, this study worked at the municipal scale (LAU2 in the EU nomenclature) and then aggregated data according to sub-regional typologies of rural areas.

A series of studies conclude that the rural development measures are less "rural" than expected due to a limited redistributive effect from the urban areas to the rural ones [29,31,33]. This study goes beyond the urban–rural dichotomy and proves the existence of significant disparities, in the distribution of policy support to investments, within the same "rural" category (between lagging and more dynamic rural areas). The distributive effects of RDP investment support measures are evidently uneven, especially in measures addressed towards agricultural and agro-industrial competitiveness, which are mainly allocated in areas already dynamic and strongly competitive. This is especially evident in North Italy, where the agri-food system is more developed than in other Italian regions. When policy instruments are governed with mechanisms based on a logic of open competition, dynamic areas are more ready than others to exploit the financial opportunities of the RDPs. In these cases, tenders boost disparities. The two territorial definitions of rural areas (from RDP and IAS) provide similar results, but the distributive effects in the inner areas are even more uneven.

Delivery mechanisms contribute to boosting disparities when funds are allocated through the "open competition" approach. EU regulations establish that open access to funds should be ensured to all potential beneficiaries, without any limitation to the competition. Still, the territorial cohesion objective added to the Lisbon Treaty does suggest the need to facilitate access to the most disadvantaged areas. On the contrary, the concept of open competition is applied in a high number of tenders, especially for policy measures aiming to strengthen agri-food competitiveness. In fact, we highlighted that the mechanism of "limited competition" (more favourable eligibility and selection criteria for lagging areas) is applied only to some categories of investment support and not to the competitiveness of farming and agri-food systems. In our analysis, some typologies of investments address lagging areas (infrastructures, small-scale social services, non-productive investments, etc.), whereas the productive investments are implicitly left to the most capable areas in terms of project design and investment rentability. This dichotomic choice seems to contradict the same officially declared strategy based on the analysis of needs in many rural development programmes, where high-quality agri-food systems in lagging areas are addressed as objective of public support. A more coherent strategy for lagging areas would have meant extending the targeting through eligible criteria exclusively addressing these areas.

## 5. Conclusions: Implications for Research and Policy Design

This study might provide two significant contributions to the methodological debate on rural areas. The first one concerns the definition of rural areas for policy purposes. When we consider the range of possible approaches to rural diversity, Italian rural policies have chosen two different approaches: in the context of the rural development policy, the Italian authorities adopted a "structural" definition, whereas, in the Inner Areas Strategy, they chose a definition closer to a "locational" approach (based on the accessibility to centres of services' provision). Based on the results of this analysis, we can conclude that both definitions have pros and cons.

The IAS definition allowed for the better identification of lagging areas and more precise quantification of support to these areas. On the other hand, the RDP definition also included agricultural variables among the criteria, which allowed us to understand how the

policy support might address different types of rural systems (intensive or marginal) and go beyond the classical dichotomic comparison between rural and non-rural. The conclusion is that a "combined" approach (mixing "structural" and "locational") might provide a better analytical framework, in line with the evolution of the literature on rurality. Furthermore, the current Italian definitions of rurality do not cover another critical group of factors explaining the characteristics of the demand for policy in the different rural areas. For example, depopulation and land abandonment have occurred in lagging areas and affected the composition and nature of the demand for policy instruments. This group of factors should be treated in more depth. In advancing this research project, there is a need to explore other types of classification of rural areas, less centred on administrative definitions.

The second relevant methodological implication relates to the long-standing debate on the effectiveness of territorial policies [34] and the contraposition between place-based and spatially blind (or generalised) policies. More recently, this debate converged upon the need for combining different types of policies under a common territorial approach [35], especially in lagging rural areas [36,37]. In this respect, the delivery systems that put more emphasis on territorial targets, as they were presented in this study, might be an essential component of a place-based policy insofar as different policy measures are combined at the correct scale [26,37].

As regards the policy implications, this research proves that in each evaluation process, the territorial distribution of public support should be the preliminary step to understanding the potential impact on the farming and rural socio-economic system. The diverse typologies of rural areas are helpful in this exercise, and analysis should always disentangle the characteristics of the delivery system. However, most evaluation reports, officially approved by monitoring committees, seldom present a sound analysis of the territorial distribution of rural development funds (and, more generally, common agricultural policy subsidies). This study has been conducted outside the formal methodological approaches promoted by the European Commission in its evaluation guidelines of RDPs.

Thus, a recommendation to policymakers and policy evaluators is to give more attention to the effects which the delivery system may generate on the distribution of public funds. This recommendation would require including this analysis in the official evaluation procedures and guidelines for the evaluation reporting activities. The analysis of the territorial distribution of public funds may also improve understanding the coherence between rural development investments and other EU policies that address territorial disparities, as pointed out by the European Court of Auditors in a recent briefing paper [38].

Territorial analysis assumes a high practical value and usability for policy design purposes. The modulation of territorial criteria by the programme regional authority in the implementation phase can provide effective results in terms of reducing disparities in funds allocation and outreaching the most lagging areas. A further policy recommendation concerns a more extensive use of territorial modulation by regional policymakers in designing the call for tenders. This recommendation can be generalised to all countries with deep territorial disparities among typologies of rural areas.

This study also assumes relevance in the context of the future CAP Strategic Plan (CSP) design [39]. Modulation of the delivery mechanisms is a key aspect of policy flexibility, especially in those countries where rural development instruments need to be decentralised under regional responsibility (i.e., Italy, Spain, Belgium, Germany, and Austria). In these countries, the socio-economic characteristics of rural areas are often very diverse from north to south and from region to region. Modulation at the regional level then becomes then a point of strength of the RDP implementation. The new regulation, imposing a unique strategic plan for the whole country, risks weakening/reducing the room for manoeuvre at the regional level by centralising the definition of implementation rules, unless the delivery mechanisms are conceived with a high degree of flexibility. Thus, future CSPs should define very general principles and implement rules and leave to regions the definition of detailed eligibility and selection criteria within multi-level governance, where the regional level decides which delivery system mostly fits the diversity of rural areas.

Finally, future policy evaluation should also deepen the characteristics of the leader delivery system and its impact on the territorial distribution of public expenditures in the LAG areas. This would require extending the analysis of calls for tender and related approved applications of the LAG's local development plan.

**Author Contributions:** Conceptualization, F.M. and G.D.F.; methodology, F.M.; software, G.A.; validation, F.M., G.D.F. and G.A.; formal analysis, F.M.; investigation, G.A.; resources, G.D.F.; data curation, G.A.; writing—original draft preparation, F.M.; writing—review and editing, F.M., G.D.F. and G.A.; supervision, G.D.F.; project administration, G.D.F.; funding acquisition, G.D.F. All authors have read and agreed to the published version of the manuscript.

**Funding:** This research was funded by the Italian Presidency of the Council of Ministers, Department for Regional Affairs and Autonomies, under specific agreement with the National Research Council-IRET (Research Institute on Terrestrial Ecosystems). Title of the research projects: "*Public instruments support for the sustainability of agri-forestry systems*".

**Data Availability Statement:** Data have been collected by the regional websites of the Department of Agriculture and Rural Development in the 21 Italian Regions and autonomous provinces. Further data have been collected by the public archive.

**Conflicts of Interest:** The authors declare no conflict of interest. The funders had no role in the design of the study; in the collection, analyses, or interpretation of data; in the writing of the manuscript; or in the decision to publish the results.

## Notes

1 Article 110 of the Regulation (EU) No 1306/2013 of the European Parliament and of the Council of 17 December 2013.

2 Article 5 of the Regulation (EU) No 1305/2013.

3 According to the EU nomenclature, the "natural areas" include different categories, ranging from Natura 2000, High Nature Value, or more simply protected areas.

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
