# Peer review of "Analysing the Policy Delivery System and Effects on Territorial Disparities in Italy: The Mechanisms of Territorial Targeting in the EU Rural Development Programmes 2014–2020"

_land, doi:10.3390/land11111883_

Round 1

Reviewer 1 Report

General Points (see attached file for details)

1. The paper presents a useful analysis of rural devt. project applications (not of the projects themselves, e.g. outcomes) in Italy during 2014-2020, related to geographical location. It develops (or reports) a number of methodological items, e.g. rural area categories/typologies, "weighted scores", with the main novelty being project selection and analysis by "mode" of competition/targeting.

2. The Introduction is rather lengthy, and contains 4 "lists" (e.g. of research questions, "relevant conditions") which readers will find hard to memorise for possible use later in the paper. 

3. Neither the "Introduction" nor the "Materials and Methods" section (nor the final "Discussion" section) emphasise that the paper's research is about Italy in 2014-2020; the special conditions of that country (other than its multiple and heterogeneous regions) and the data period are never mentioned, so that generalisations to other places and period are left undiscussed.

4. Somewhat similarly, the paper analyses only investment subsidies (though these are not defined, in conceptual terms); this is not sufficiently stressed in the Abstract (which starts with "rural devt. measures") ir in the Discussion.

5. It is often unclear whether text refers to analysis by officialdom (e.g. formal evaluation by MAs or by national/EU authorities, who must follow predetermined approaches) or to analysis by external consultants, academics, etc., who are free (as in this paper) to determine their own methodologies. This obscures some of the paper's novelties (which are also not brought out in the Abstract or Discussion). .

6. Comments in the attached Word file often relate to obscurities in the terms used in the paper, e.g.

areas/regions (some official, some not)

tenders (by project proposers) or tender calls (by managing authorities)

7. The English is generally good, but rather verbose in places (my TCs try to reduce these), and the words "the" (referring to an undefined noun) and "relevant" (to nothing obvious) are thus mis-used in several places.

8. In my view, all tables and figures should be "introduced" in the text before they are presented, and at least some of their features "discussed" (before or after). Readers should not be faced with an indigestible item, and no explanatory text. In this paper, this is not done in all cases.

Please see attached Word version of the paper, with detailed Track Changes (mostly improvements to the English) and Comments (mostly queries as to meaning).

Reviewer 2 Report

Undoubtedly, the work presented is very interesting, given that, the topic (rural development programs in Italy) is very current.
The structure of the paper is correct, although the content could be improved and the following comments should be taken into account.

- In general, bibliographical support is almost non-existent. Although the legal basis is very important in this subject, it is impossible to carry out a research article with only 3 bibliographical references. A State of the Matter section is necessary to give scientific solvency to the research and then be debated in the Discussion section.
-
More bibliographical references on rural development policy in Europe and in Italy are necessary. There is a lot of research on it.
-         The Discussion section is not correct.
-         There is no conclusion section.
-          Regarding the methodology, it is correct and well applied, but scientifically inconsistent, since it is not correctly justified. Although it is decided to apply a descriptive methodology, it must also be supported by existing research by the scientific community. Best regards.

Reviewer 3 Report

The topic of the manuscript and the objective are really attractive and interesting. However, the authors must improve some aspects.

Firstly, the low number of references. Some Italian authors worked on similar topics of the uneven distribution of EU rural development funds: Francesco Pagliacci, Marilena Labianca and José Antonio Cañete.

Secondly, the classification of rural areas, following the definitions of the EU practice, does not more scientific and academic. Some indicators could be incomes/inhabitant or inhabitants density?. Why is it not differentiated into three categories: deep rural areas, intermediate and dynamics rural areas?.

Thirdly, the not inclusion of the LEADER approach in the analysis is doubtful. Some authors (Labianca, Cañete) proved, too, the uneven distribution of this initiative.

Fourthly, the consideration of the distribution between beneficiaries or the involvement of particular funds in these investments. 

Fifthly, what about maps? if the authors are dealing with spatial distribution, it is relevant to use the cartographic tool to show the uneven distribution of funds.

Sixthly, the authors don´t propose solutions/alternatives to this proven uneven distribution.

And finally, the discussion lacks a comparison with other studies and the main contributions of this research.

Round 2

Reviewer 2 Report

with the changes introduced the paper can be published.